# Metabolomics for predicting fetal growth restriction: protocol for a systematic review and meta-analysis

Debora Farias Batista Leite,[1,2] Aude-Claire Morillon,[3] Elias F Melo Júnior,[1] Renato T Souza,[2] Ali S Khashan,[3,4] Philip N Baker,[5] Louise C Kenny,[3,6] José Guilherme Cecatti[2]

For numbered affiliations see end of article.

**Correspondence to**
Professor José Guilherme Cecatti; cecatti@unicamp.br

## ABSTRACT

**Introduction** Fetal growth restriction (FGR) is a relevant research and clinical concern since it is related to higher risks of adverse outcomes at any period of life. Current predictive tools in pregnancy (clinical factors, ultrasound scan, placenta-related biomarkers) fail to identify the true growth-restricted fetus. However, technologies based on metabolomics have generated interesting findings and seem promising. In this systematic review, we will address diagnostic accuracy of metabolomics analyses in predicting FGR.

**Methods and analysis** Our primary outcome is small for gestational age infant, as a surrogate for FGR, defined as birth weight below the 10th centile by customised or population-based curves for gestational age. A detailed systematic literature search will be carried in electronic databases and conference abstracts, using the keywords 'fetal growth retardation', 'metabolomics', 'pregnancy' and 'screening' (and their variations). We will include original peer-reviewed articles published from 1998 to 2018, involving pregnancies of fetuses without congenital malformations; sample collection must have been performed before clinical recognition of growth impairment. If additional information is required, authors will be contacted. Reviews, case reports, cross-sectional studies, non-human research and commentaries papers will be excluded. Sample characteristics and the diagnostic accuracy data will be retrieved and analysed. If data allows, we will perform a meta-analysis.

**Ethics and dissemination** As this is a systematic review, no ethical approval is necessary. This protocol will be publicised in our institutional websites and results will be submitted for publication in a peer-reviewed journal.

**PROSPERO registration number** CRD42018089985.

## Strengths and limitations of this study

► This systematic review covers a great range of electronic databases and will also search for grey literature.
► Two researchers will perform literature search, data extraction and study quality assessment independently, and any disagreement will be resolved by a third reviewer.
► Careful statistics procedures will be performed to identify accuracy of metabolomics in predicting fetal growth restriction.

## INTRODUCTION

Fetal growth restriction (FGR) is usually defined as a fetus that has not reached its intrauterine growth potential,[1 2] with no major congenital abnormalities[1] and has also been named as fetal growth retardation, intrauterine growth restriction or retardation.[3] This heterogeneous condition is associated with increased risks of stillbirth,[4 5] neonatal intensive care unit admission,[6] neonatal mortality,[5]

cognitive and behavioural impairment in infancy[7] and chronic non-transmissible disease in adulthood.[8] FGR is mainly diagnosed according to the estimated fetal weight in ultrasound scans below the 10th centile,[2 9] although it is anticipated that misdiagnosis can occur: fetuses below the 10th centile, but with normal outcomes ('constitutionally' small), or fetuses above the 10th centile, but who did not follow personal growth potential.[2] In this context, antenatal recognition of truly restricted fetuses, that is, those at higher risk of morbidity and mortality in any period of life, followed by adequate obstetrical care, can improve neonatal outcomes.[10]

Unfortunately, in current practice, there is no gold standard for FGR diagnosis. Recent consensus has added ultrasound criteria (eg, abdominal circumference, umbilical and uterine artery Doppler measurements) and lowered estimated fetal weight cut-offs (<3rd centile),[1] to improve specificity. In these terms, the concept of FGR can overlap with that of small for gestational age (SGA), which includes infants with birth weight below the 10th (or fifth, or third) centile for gender and gestational age.[11] In fact, it is common to use SGA as a surrogate for FGR,[6 12 13] as an indication of real intrauterine growth impairment. Besides that, neonatal parameters seem more

adequate as 'patient important outcomes', but regrettably, ultrasound have still low accuracy to determine them.[6]

Since 1990s, when the thrifty phenotype theory was introduced,[14] a huge effort has been undertaken to investigate the pathologically growth restricted fetuses and newborns and to enhance antenatal screening.[6] Clinical data has been intensively studied, with conflicting risk factors[15 16] and, in general, poor accuracy is achieved for identifying impaired birth weight[17] or neonatal morbidity,[6] even when first,[18] second[13 17] or third trimester[19] ultrasound parameters are added to prediction model. Using only clinical or ultrasound variables, the great majority of SGA babies will only be recognised after birth, by population-based[5] or customised curves.[17] Biomarkers, such as placental growth factor (PlGF), soluble fms-like tyrosine kynase 1 (s-Flt-1) and alpha-fetoprotein,[20] have each been found to show promise as aids to understanding FGR. However, the performance of these angiogenic factors as predictors of FGR has been limited (positive likelihood ratio, LR+, of 1.3 for PlGF and 1.4 for s-Flt-1).[21] Similarly, placental proteins are not robust enough biomarkers for FGR (eg, LR+ of 3.7 for placental protein-13 in first trimester).[22] Therefore, there is a real need for better methods of FGR prediction.

The disappointing evidence may be due to the multifactorial nature of FGR; the aetiology of the condition is complex and poorly defined. Moreover, placental structure and functioning, maternal and fetal metabolism vary during pregnancy.[23] In this context, contemporary metabolomics approaches have identified several pathways and metabolic processes that may contribute to FGR, such as disruptions in DNA methylation,[24] cellular signalling,[25 26] neurotransmitter precursors[26 27] and energy generation.[25 26]

Despite excellent performance of some metabolites in predicting FGR (area under the curve, above 0.9),[25 26] these studies have shown an overall modest accuracy.[21] However, only two 'omics' studies were included in Conde-Agudelo *et al*[21] review, and issues related to gestational age of sampling and delivery, or analysis of composite outcomes, could have introduced bias and confounders to metabolomics findings. In recent years, many authors have applied diverse metabolomics techniques to predict FGR, suggesting that metabolite biomarkers may have a role to play in disease screening. Thus, the main objective of this systematic review is to define the accuracy of metabolomics techniques for predicting FGR. As secondary aims, we will try to determine which metabolites are robust candidates for a prediction model of FGR and which chemical class they belong to.

## METHODS AND ANALYSIS
### Review question
What is the accuracy of metabolomics for predicting FGR?

### Condition or domain studied
SGA infant and FGR.

### Participants/population
Inclusion criteria: Original studies including pregnant women.
Exclusion: Congenital malformation.

### Interventions/exposure
Screening for SGA/FGR with metabolomics approach. Biomarker analysis should have been performed on samples taken before clinical recognition of neonatal outcome.

### Inclusion and exclusion criteria
Original studies (cohort or case control studies) involving pregnant women, as the studied population and SGA infant (and variations of terminology), as the outcome of interest, will be included in this systematic review.

The reasons for excluding studies are: (1) if they are Cross-sectional studies, Case Reports, Editorials, Letter to Editors, Commentaries, Expert Opinions, or any type of Reviews; (2) if they describe only experimental studies with animals; (3) if they show duplicate publication of the same data; in these cases, we will use the most recent publication.

### Outcomes
#### Primary outcome
SGA infant, defined as a birth weight below the 10th centile according to population-based or to customised charts.

#### Secondary outcomes
Birth weight below the fifth or the third centile by population-based or customised parameters.

### Literature search
The primary source of information will be these electronic databases: PubMed, EMBASE, Latin American and Caribbean Health Sciences Literature, Scientific Electronic Library Online, Health Technology Assessment, Database of Abstracts of Reviews of Effects, Aggressive Research Intelligence Facility, Cumulative Index of Nursing and Allied Health Literature, Maternity and Infant Care, Scopus and Web of Science. Secondary sources include Google Scholar, hand-held searching of the reference list of eligible studies, conference proceedings and contact with authors when necessary.

The keywords linked to the outcomes of interest will be combined with terms related to 'metabolomics' technique, 'pregnancy' and 'screening', using Boolean connectors. The same search strategy will be applied for each database, adapting for individual filters, main language, their own syntax and mechanisms of search; the complete search strategy is provided as online supplementary material.

Considering that the term metabolome was first used in 1998,[28] we will take into account studies published in

the last 20 years (1998–2018). The preliminary searches for this systematic review have started in February 2018. The search strategy will be re-run before final analysis, to check for recently published eligible studies. There are no language restrictions.

### Data extraction and management

All searches will be exported to a reference manager (EndNote). Individually, two researchers (DFBL and ACM) will select papers according to (1) title or abstract and (2) full text, that will be read only when abstracts are not sufficient to decide about inclusion criteria. Any disagreement about selected studies will be dealt by a third researcher (EFMJ or RTS); in these cases, only after majority decision (2:1 ratio) the next step will be performed. A fifth investigator (JGC) will revise all procedures before approving the data extraction. DFBL, ACM and ASK will deal with the statistic procedures. JGC, PNB and LCK will re-examine all steps and supervise data interpretation.

A standardised form will be applied to extract the variables of interest—by two independent researchers—which will include: authors and year of publication, country of participants' enrolment, study design, definition used for FGR/SGA (customised or population-based charts) and outcome measured, number of affected (who later delivered a FGR/SGA baby) and non-affected pregnant women, gestational age of assessment (throughout pregnancy), laboratory methods and biological sample analysed (eg, blood, amniotic fluid). In addition, data regarding growth impairment suspicion in pregnancy—such as gestational age, criteria applied for diagnosis and follow-up—will be retrieved once available. Researchers will contact authors (by electronic address) if any clarification of data is needed. The metabolites described will be matched with the Human Metabolome database to check their characteristics.[29]

### Strategy for data synthesis

Details about data search and selection will be presented as a flow diagram, according to the Preferred Reporting Items for Systematic Reviews and Meta-Analyses (PRISMA) statement recommendations.[30] An aggregate participant data synthesis will be performed with all included studies; narrative data will be analysed and structured according to birth weight centile (10th, 5th and 3rd) and curve type (population-based or customised curves). Additionally, the metabolites will be grouped and synthesised according to their biological function and chemical subclass. Studies' characteristics and risk of bias assessment will be demonstrated in tables. Once possible, we will perform subgroup analysis according to:

► Which metabolomic methods were applied (gas or liquid chromatography coupled with mass spectrometry; or proton nuclear magnetic resonance).
► Maternal health status before pregnancy (healthy ones vs women with any chronic health condition).

► Gestational age of first fetal growth impairment suspicion (early vs late FGR).[1]
► Type of pregnancy (single vs multiple).

Depending on data availability, accuracy measures will be calculated and a meta-analysis will be drawn. Considering the quantitative nature of the metabolomics approach and the expected different thresholds for metabolites in each study, we will try to perform the hierarchical summary receiver characteristic operating curve.[31] Heterogeneity will also be assessed, through $I^2$ test.

### Risk of bias assessment

Both investigators initially involved with literature search (DFBL and ACM) will assess methodological quality and applicability of all included studies, and they must check their judgements. A third researcher (EFMJ or RTS) will resolve any disagreement if necessary and the final decisions will be made by majority. We will use the 'Quality Assessment of Diagnostic Accuracy Studies'[32] tool, which comprised four domains: patient selection, characteristics of index test (metabolomics technique), the reference standard test (measurement of birth weight) and flow and timing of patient inclusion and follow-up. Every study will be labelled as 'low', 'high' or 'unclear' risk of bias for each domain. For example, there is 'low risk of bias' if the study clearly states how the metabolomics techniques were performed, or which birth weight curve was applied to identify the SGA babies.

Regarding publication bias, we will assess the symmetry of funnel plots if more than ten studies are included in the meta-analysis.[33]

### Potential limitations to this review

Concerning the publication bias, we expect to encounter more published positive results and data interpretation must take this issue in consideration. The metabolomics approach is highly detailed and meticulous, has shown great technological advancements in recent years, and results from mass spectrometry and from nuclear magnetic resonance complement each other. Therefore, we acknowledge that we may find distinct metabolites in each study and generalisation may be challenging.

In this systematic review, we have considered SGA as a proxy for FGR, as other authors.[6 12 13] The consensus for FGR diagnosis was published recently[1] and past investigations may have used distinct terminology or conflicting criteria for this condition in pregnancy. Additional confounders to interpret the selected studies will include clinical factors potentially associated to FGR/SGA, like parity, smoking habits and history of previous fetal growth impairment. These characteristics will be appraised during data extraction and synthesis, and detailed evidence will be retrieved.

### Ethics and dissemination

This protocol follows the PRISMA Protocols statements.[34] A report of this systematic review will be sent to our

sponsors. This protocol will be electronically available on UNICAMP-CNPq-Gates Foundation project website (www.medscinet.com/samba) and Infant Centre website (infantcentre.ie). Our results will be submitted to publication in peer-reviewed journal.

## Patient and public involvement

Patients and or public were not involved at all in elaborating this systematic review protocol.

## CONCLUSION

This systematic review will synthesise data about metabolomics and FGR/SGA, a promising field for understanding disease pathophysiology and natural history. By highlighting the metabolites and chemical classes that they belong to, this review might present solid data to future research protocols, that can target the most promising compounds, or assess the participants in a more reliable gestational age, for example. A robust FGR/SGA prediction assumes great importance in reproductive health and epidemiology, since this condition is associated with short and long-term adverse outcomes for the offspring.

#### Author affiliations
[1]Department of Maternal and Child Health, Clinics Hospital of Federal University of Pernambuco, Recife, Brazil
[2]Department of Gynaecology and Obstetrics, University Campinas, Sao Paulo, Brazil
[3]Irish Centre for Fetal and Neonatal Translational Research (INFANT), University College Cork, Cork, Ireland
[4]School of Public Health, University College Cork, Cork, Ireland
[5]College of Life Sciences, University of Leicester, Leicester, UK
[6]Department of Women's and Children's Health, Faculty of Health and Life Sciences, Institute of Translational Medicine, University of Liverpool, Liverpool, UK

**Acknowledgements** We would like to thank Shauna Barret, librarian of Brookfield Library, University College Cork, Ireland, for her support with the literature search strategy. We are also grateful to Maeve O'Connell, for her suggestions to the final manuscript.

**Contributors** DFBL (the guarantor of the review) and A-CM developed the systematic review protocol and will perform the literature search, study selection, data extraction and risk of bias assessment. ASK, PNB, RTS, EFMJ and JGC supervised protocol elaboration and the latter three will resolve any discrepancy about methodology. ASK, DFBL and A-CM will deal with statistics procedures. PNB and LCK performed the last amendments of protocol and will revise the final systematic review draft. All authors have read this manuscript and have agreed with this submission.

**Funding** This research was supported by Brazilian National Research Council (grant number 401636/2013-5) and Bill and Melinda Gates Foundation (grant number OPP1107597-Grand Challenges Brazil: Reducing the burden of preterm birth, Fiotec number 05/2013), which provided funding to PRETERM-SAMBA project (www.medscinet.com/samba). DFBL was granted a scholarship (process number 88881.134512/2016-01) from the Brazilian Federal Agency for Support and Evaluation of Graduate Education (CAPES) and developed part of her doctoral studies as a visiting student at University College Cork, Ireland. RTS has also awarded a scholarship from CAPES (process number 88881.134095/2016-01). A-CM was granted a scholarship from Science Foundation Ireland, for her doctoral thesis. Our sponsors have not intervened in authors' decision to write the systematic review protocol or to submit this paper.

**Competing interests** DFBL and A-CM are studying this technology in predicting FGR. JGC, LCK and PNB have presented conference talks about this field. LCK and PNB are principal investigators of Metabolomic Diagnostics.

**Patient consent** Not required.

**Ethics approval** As this is a systematic review protocol, no ethics committee approval is necessary.

**Provenance and peer review** Not commissioned; externally peer reviewed.

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
