## [Reviewer comments · BMJ Open]

ARTICLE DETAILS

TITLE (PROVISIONAL)	Metabolomics for predicting fetal growth restriction: protocol for a systematic review and meta-analysis
AUTHORS	Leite, Debora; Morillon, Aude-Claire; Melo Júnior, Elias; Souza, Renato; Khashan, Ali; Baker, Philip; Kenny, Louise; Cecatti, Jose Guilherme

VERSION 1 – REVIEW

REVIEWER	Natalija Vedmedovska Riga Stradins University, Riga, Latvia
REVIEW RETURNED	04-May-2018

GENERAL COMMENTS	In the abstract "Fetal growth retardation" was mentioned as a keywords, but it did not match to the keywords in the main manuscript, and this term not in use anymore. I missed in the section "Risk of bias assessment" the possible estimation of likely size of the publication bias in the review and an approach/methods to dealing with this bias.
---

REVIEWER	Alexandros A Moraitis University of Cambridge, UK
REVIEW RETURNED	03-Jul-2018

GENERAL COMMENTS	This systematic review is clinically important and the protocol was well designed. The statistical methods are as expected for a systematic review of predictive accuracy. Some minor comments: 1) In the methods section the authors need to specify if they are going to use samples from a specific trimester or throughout the pregnancy. 2) In the methods section the authors need to specify if they are planning to do the data synthesis for each metabolite separately or they are going to use combinations. If they plan the latter, can they pre-specify any combinations? 3) The authors need to explain why they have used 1998 as the first year of their review. I understand that metabolomic research before that year is unlikely but why didn't they use for example 1996 or 2000? Saying that they included studies published in the last 20 years is not good enough. A potential explanation would be that the first paper on that topic was published in 1998.
---

REVIEWER	Wessel Ganzevoort Amsterdam UMC, Amsterdam, The Netherlands
REVIEW RETURNED	18-Jul-2018

GENERAL COMMENTS	I like the subject, it is an important upcoming field which may yield new valuable biomarkers for an important condition. The author group holds renowned authors. The overall approach appears solid, and I am sure it will become a fruitful SR. Two minor points: In their introduction the authors consider the fact that the term SGA is not the topic of interest but FGR is. Yet, they should explain a little bit better why they chose SGA as the matter of outcome. If they chose SGA (just the statistical deviation of size), they should refer to it as SGA. As long as we keep on mixing the terms we will keep on having disappointing results (page 7 Line 22) because associations are not seen because an FGR population is diluted with healthy SGA babies. So please be careful with nomenclature. And please refer to this issue in the limitations of the study in the discussion. I would suggest to also focus at the difference in predicting late and early FGR as a secondary analysis, as they may have a different pathophysiology, different phenotype and very different clinical impact.
---

VERSION 1 – AUTHOR RESPONSE

Firstly, regarding the Editor’s comments, we agree that public and patients can be involved in the design and planning of a systematic review. However, they have not participated in this protocol; this amendment is on Page 10. The PRISMA-P checklist is uploaded at the BMJOpen (ScholarOne Manuscripts™) submission webpage. Our sponsors have had no role in deciding to perform or to submit this protocol for publication (Page 14, lines 20-22).

Secondly, we comprehend Prof Natalija Vedmedovska (1st reviewer) care with publication bias assessment, and we will perform the Deek’s test for those studies included in the metaanalysis (Page 9, lines 8-9). Although ‘fetal growth restriction’ is a more common terminology, it is not listed as a Medical Subject Heading (MeSH), while ‘fetal growth retardation’ is still considered for the PubMed database controlled vocabulary. Thus, we hypothesize we would magnify the dissemination of metabolomics and fetal growth restriction knowledge by adding this keyword.

Thirdly, we apologize Prof Alexandros A. Moraitis (2nd reviewer) for any doubts regarding the data extraction description. We will consider samples collected throughout pregnancy (Page 8, 1st line), and

will synthesize metabolites' data according to their biological function and chemical subclass (Page 8, lines 15-16). We have chosen 1998 as the starting date for the literature search since the terminology 'metabolome' was first used at that time (Page 7, lines 16-17).

Finally, we understand Prof Wessel Ganzevoort (3rd reviewer) concerns about terminology misunderstandings (fetal growth restriction, FGR, *versus* small for gestational age, SGA). Unfortunately, the current tools (clinical examination, laboratorial analysis, ultrasound scans) fail to predict babies at risk of adverse outcomes (Page 4, lines 20-22; Page 5, lines 1-2). Conversely, the birthweight seems a more reliable tool to identify these newborns, and we suspect that is probably the reason why many authors use SGA as a surrogate for FGR. Thus, we have chosen birthweight centiles as the outcome. We hope future studies could investigate the prediction of FGR using the new consensus-based criteria (Page 9, lines 18-24). We also agree that early and late FGR have different natural histories and clinical impact. Once available, we will extract and synthesize data about of growth impairment suspicion during pregnancy (Page 8, lines 2-4, and 22).

In addition, we presume that single and multiple pregnancies affected by fetal growth restriction may have a different pathophysiology and clinical course. This is the reason why we have added this subgroup analysis (Page 6, lines 12 and 20; Page 8, line 23).

VERSION 2 – REVIEW

REVIEWER	Alexandros Moraitis University of Cambridge
REVIEW RETURNED	24-Aug-2018
GENERAL COMMENTS	I'm happy with the minor amendments. No further revision required.